# Dependency-Aware Semi-Structured Sparsity of GLU Variants in Large Language Models

**Zhiyu Guo**                                                            *guo.zhiyu.fy1@is.naist.jp*
*Nara Institute of Science and Technology*

**Hidetaka Kamigaito**                                                   *kamigaito.h@is.naist.jp*
*Nara Institute of Science and Technology*

**Taro Watanabe**                                                        *taro@is.naist.jp*
*Nara Institute of Science and Technology*

**Reviewed on OpenReview:** *https://openreview.net/forum?id=T5OuTgPxHS*

## Abstract

The rapid advancement in Large Language Models (LLMs) has markedly enhanced the capabilities of language understanding and generation. However, the substantial model size poses hardware challenges, affecting both memory size for serving and inference latency for token generation. To address those challenges, we propose Dependency-aware Semi-structured Sparsity (DaSS), a new method for the recent prevalent GLU-based LLMs pruning, which incorporates structural dependency into the weight magnitude-based unstructured pruning. We introduce an MLP-specific pruning metric that evaluates the importance of each weight by jointly considering its magnitude and its corresponding MLP intermediate activation norms. DaSS facilitates a balance between the adaptability offered by unstructured pruning and the structural consistency inherent in dependency-based structured pruning. Empirical evaluations on LLaMA2, Mistral, and Gemma model families demonstrate that DaSS not only achieves superior perplexity and accuracy compared to SparseGPT and Wanda in achieving hardware-friendly N:M sparsity patterns but also maintains the computational efficiency of Wanda. Code is available at `https://github.com/guozhiyu/glu_dass`

## 1 Introduction

Recent years have witnessed the great success of Large Language Models (LLMs) across various challenging tasks, such as mathematical reasoning, code generation. However, the practical use of these models for inference has faced a major obstacle due to the substantial computational resources they consume. To tackle this, many of the key developments up to now have revolved around weight quantization. It is possible to quantize LLMs down to 4 bits per weight with little impact on accuracy, which aids in memory reduction and speeds up inference (Lin et al., 2024). Nonetheless, maintaining accuracy becomes problematic when quantizing to around 3 bits per weight with existing methods (Dettmers et al., 2024; Egiazarian et al., 2024).

A complementary method is neural network pruning (Han et al., 2015b), which can be combined with quantization to further improve the inference efficiency of LLMs (Kurtic et al., 2023; Frantar & Alistarh, 2023). Pruning can be categorized into two main approaches: unstructured pruning (Sun et al., 2024; Frantar & Alistarh, 2023), which involves the removal of specific weights, and structured pruning (Ma et al., 2023), which entails the removal of complete rows or columns of weights. In contrast to structured pruning, which struggles with performance in LLMs even at low sparsity levels, unstructured pruning methods like SparseGPT (Frantar & Alistarh, 2023) and Wanda (Sun et al., 2024) exhibit promising results without additional retraining, and achieves practical speedup in both CPU and GPU through the recent engineering advancements (Agarwalla et al., 2024). They also have the benefit in reducing hallucinations of LLMs (Chrysostomou et al., 2024).

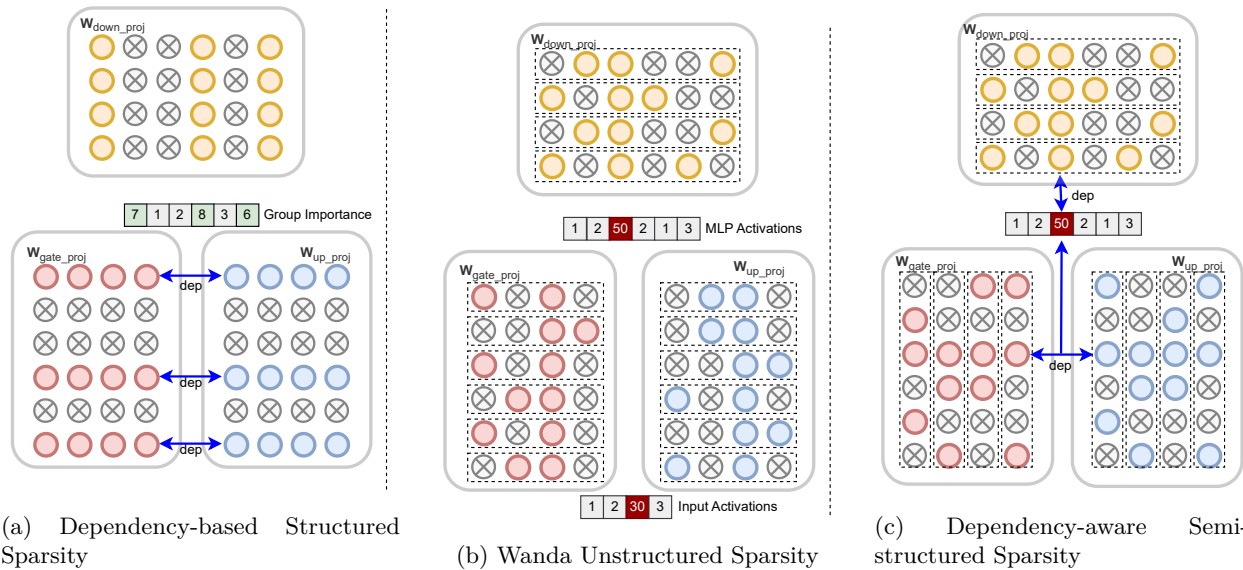

Figure 1: Illustration of Dependency-aware Semi-structured Sparsity (DaSS). In (a) dependency-based structured pruning (Ma et al., 2023), all the weights connecting to the same intermediate neuron are removed or remain simultaneously. In (b) Wanda unstructured pruning (Sun et al., 2024), it assigns greater emphasis to the weights corresponding to large input activations. For `Gate-Proj` and `UP-Proj`, the same number of weights are removed for each MLP neuron, regardless of whether some neurons have much larger activation norms. For `Down-Proj`, the weights corresponding to larger activation norms are more likely to be pruned. This can lead to a structural mismatch. In (c), all the weights corresponding to large intermediate activations are more likely to be reserved.

Modern LLMs, such as LLaMA2 (Touvron et al., 2023) and Mistral (Jiang et al., 2023), have adopted several architectural changes, including the use of GLU (Gated Linear Unit) variants (e.g., SwiGLU) (Shazeer, 2020) in the MLP modules, and grouped-query attention (GQA) (Ainslie et al., 2023). As the GLU-based MLP module accounts for more than 80% of the parameters in LLMs that use GQA [1], its pruning emerges as a pivotal factor in determining the overall compression efficacy of LLMs. In dependency-aware structured pruning, it's critical to consider that pruned parameters have dependencies with other parameters, owing to their interconnected nature (Ma et al., 2023; Fang et al., 2023). In the context of MLP pruning, all the weights connected to each intermediate neuron should be preserved or pruned simultaneously. The precise coordination involved in pruning is crucial for upholding the model's structural integrity and its functional capabilities. Although current unstructured pruning methods (Sun et al., 2024; Frantar & Alistarh, 2023) effectively remove a significant number of redundant weights, they operate entirely locally within each linear layer without considering inter-dependencies to other layers. This can lead to a structural mismatch, which is more evident in Wanda as shown in Figure 1b: In Gate and Up projections, the same amount of parameters are pruned for each MLP neuron. However, the intermediate activation norms of GLU are not uniformly distributed and some neurons have much larger norms than others. Based on Wanda pruning metric, more weights connected to neurons with large activation norms are preserved. At high sparsity, this problem gets magnified in Wanda causing significant drops in performance by the broken network.

In order to overcome the limitations present in current pruning methodologies, we introduce a new paradigm, namely, **D**ependency-**a**ware **S**emi-structured **S**parsity (DaSS). This approach is specifically designed to navigate the middle ground between the flexibility of unstructured pruning and the structural consistency of dependency-based structured pruning. To emphasize the importance of weights corresponding to large intermediate activations, we present a new MLP pruning metric that assesses each weight's importance based on the product of its magnitude and the norm of the corresponding MLP intermediate activations. Our

---

[1]In LLaMA2-70B, the dimension of key and value is $\frac{1}{8}d$, MLP intermediate dimension is $\frac{7}{2}d$.

proposed DaSS method, illustrated in Figure 1c, embodies a semi-structured pattern that retains a degree of the adaptability inherent in unstructured pruning while incorporating the dependency-aware aspect of structured pruning. DaSS refines Wanda's pruning criterion by replacing input activation norms with those of the intermediate activations generated within the MLP. Specifically, while Wanda's importance score for a weight is the product of its magnitude and the corresponding input activation norm, DaSS uses the product of the weight magnitude and the norm of the corresponding intermediate activation. This subtle yet significant alteration facilitates a more comprehensive pruning strategy that accounts for the structural dependencies within the MLP. DaSS can be easily extended to hardware-friendly N:M sparsity patterns Mishra et al. (2021).

We perform extensive experiments on LLaMA2 (Touvron et al., 2023), Gemma (Team et al., 2024), and Mistral (Jiang et al., 2023) to evaluate DaSS across various tasks from language modeling, 5 commonsense reasoning tasks. In achieving hardware-friendly N:M sparsity patterns, DaSS consistently surpasses SparseGPT and Wanda in terms of perplexity and accuracy[2], while preserving the computational efficiency of Wanda. Moreover, DaSS demonstrates consistent effectiveness in all the prevalent GLU variants, including SwiGLU, GeGLU and ReGLU. Impressively, DaSS outperforms SparseGPT at high sparsity even without weight update. However, realizing the full performance benefits of this aspect is currently constrained by the nascent state of N:M sparsity support in libraries such as PyTorch, particularly for the input-balanced pruning needed in the `Gate-Proj` and `Up-Proj` components. We anticipate these limitations, discussed further in the Limitations section, will be addressed in future library updates, unlocking the full potential of DaSS's N:M sparsity implementation. We hope our fresh insights can motivate more nuanced GLU-specific LLM compression strategies.

## 2 Preliminaries

### 2.1 Wanda Pruning Method

In the context of LLM pruning, we denote a linear layer weight matrix $\mathbf{W} \in \mathbb{R}^{d_{\text{out}} \times d_{\text{in}}}$ and input activations $\mathbf{X} \in \mathbb{R}^{L \times d_{\text{in}}}$, where $d_{\text{in}}$, $d_{\text{out}}$ and $L$ represent weight input dimension, weight output dimension, and input sequence length, respectively. Wanda (Sun et al., 2024) evaluates weight importance through the multiplication of the weight magnitude with the norm of the input feature. Specifically, the importance score for a given weight $W_{i,j}$ is determined as follows:

$$I_{i,j} = |W_{i,j}| \cdot \|\mathbf{X}_j\|_2 \tag{1}$$

Here, $I_{i,j}$ is the weight importance score for a given weight $W_{i,j}$ in the weight matrix. $\|\mathbf{X}_j\|_2$ represents the $\ell_2$ norm of the $j^{th}$ feature of input $\mathbf{X}$, which is calculated across all $L$ tokens to produce a scalar. In Wanda pruning, importance scores for weights are compared for each output individually, corresponding to each row in the matrix $\mathbf{W}$. We call this as *output-balanced* granularity.

### 2.2 GLU Variants for Transformer

Gated Linear Units (GLU) (Dauphin et al., 2017) are formed by the element-wise multiplication of two linear projections, with a sigmoid function applied to one projection before the multiplication. Shazeer (2020) suggests an alternative design for the Transformer's MLP layer that incorporates GLU variants, effectively replacing the conventional first linear transformation and activation function. More formally, we denote the $d_{\text{hidden}}$ as the dimension of the model's hidden state and $d_{\text{int}}$ as intermediate dimension of the MLP module. For $\mathbf{x} \in \mathbb{R}^{d_{\text{hidden}}}$, $\mathbf{W^{(1)}} \in \mathbb{R}^{d_{\text{hidden}} \times d_{\text{int}}}$, $\mathbf{W^{(2)}} \in \mathbb{R}^{d_{\text{hidden}} \times d_{\text{int}}}$, $\mathbf{W^{(3)}} \in \mathbb{R}^{d_{\text{int}} \times d_{\text{hidden}}}$, and activation function $\sigma(\cdot)$, each MLP layer produces $\mathbf{z} \in \mathbb{R}^{d_{\text{hidden}}}$ by evaluating

$$\mathbf{y} = \sigma(\mathbf{x}\mathbf{W^{(1)}}) \otimes \mathbf{x}\mathbf{W^{(2)}} \tag{2}$$

$$\mathbf{z} = \mathbf{y}\mathbf{W^{(3)}} \tag{3}$$

---

[2]It is common in unstructured pruning research that different methods achieving the same sparsity level exhibit similar inference speeds. Our work, like others in this area, focuses on enhancing accuracy at a given sparsity.

We call $\mathbf{W^{(1)}}, \mathbf{W^{(2)}}, \mathbf{W^{(3)}}$ linear projections as `Gate-Proj`, `Up-Proj`, and `Down-Proj`, respectively. GLU variants that use Swish (Ramachandran et al., 2017), GeLU (Hendrycks & Gimpel, 2016), and ReLU (Glorot et al., 2011)activation functions in Equation 2 are called SwiGLU, GeGLU, and ReGLU, respectively. SwiGLU is most widely used in the recent LLMs.

### 2.3 Output-balanced Pruning Limitations

N:M sparsity pattern offers notable speed improvements on recent NVIDIA GPUs (Mishra et al., 2021; Kurtic et al., 2023). It is specified that every group of M consecutive weights must include N zeros. SparseGPT (Frantar & Alistarh, 2023) is not explicitly designed for output-balanced pruning granularity. When converting into an N:M sparsity pattern, SparseGPT forces every M consecutive weight in each row to have exactly N zeros. In such cases, SparseGPT (Frantar & Alistarh, 2023) and Wanda (Sun et al., 2024) unanimously remove the same amount of weights for each output. Such methods often emphasize the significance of individual components within the weight matrix and overlook inter-dependencies to other layers in the network. For MLP input projections pruning, an equal amount of weights corresponding to each intermediate neuron are removed. However, for output projection based on SparseGPT and Wanda pruning metrics, the weights connecting to intermediate neurons with larger activations are more likely to be reserved, leading to a structural mismatch.

## 3 Dependency-aware Semi-structured Sparsity

In this section, we introduce **D**ependency-**a**ware **S**emi-structured **S**parsity (DaSS) for pruning MLP, which incorporates structural dependency into weight magnitude-based unstructured pruning method. An overview of DaSS and its comparison with the existing pruning method is shown in Figure 1.

Here, we denote the transposed weight matrices $\mathbf{W^{(1)}}, \mathbf{W^{(2)}}, \mathbf{W^{(3)}}$ in Equations 2 and 3 as $\mathbf{W^{(1)\top}}$, $\mathbf{W^{(2)\top}}$ and $\mathbf{W^{(3)\top}}$, respectively. For $\mathbf{W^{(1)\top}}$ and $\mathbf{W^{(2)\top}}$, the shapes are $(d_{\text{int}}, d_{\text{hidden}})$. For $\mathbf{W^{(3)\top}}$, the shape is $(d_{\text{hidden}}, d_{\text{int}})$.

**Structure Dependency in MLP.** In dependency-based structured pruning (Ma et al., 2023; Fang et al., 2023), the initial step is dedicated to recognizing groups of interconnected structures within the model. In terms of GLU-based MLP module pruning, there are three projection matrices, all weights connecting to an identical intermediate neuron are collectively classified into the same dependency group. When pruning weights in $\mathbf{W_{i,:}^{(1)\top}}$, it is essential to also prune all the weights in $\mathbf{W_{i,:}^{(2)\top}}$ and $\mathbf{W_{:,i}^{(3)\top}}$ to maintain the structural consistency of the pruned neural network. In DaSS pruning, instead of aggressively pruning all the weights in less important dependency groups, we incorporate such structural coordination into unstructured pruning. If all weights in $\mathbf{W_{:,i}^{(3)\top}}$ are emphasized importance through additional importance indicator, such as augmenting corresponding input activations in Eq. 1, then the weights in $\mathbf{W_{i,:}^{(1)\top}}$ and $\mathbf{W_{i,:}^{(2)\top}}$ should also be emphasized similar importance.

**Incorporating Dependency into Weight Importance.** In dependency-based pruning, we assess the importance of each weight and then aggregate the importance scores within the same group as the group importance score. Consequently, each weight within the same dependency group shares a consistent importance score, ensuring their simultaneous retention or elimination. The importance score of each weight is equal to the group importance score. In DaSS pruning, we take both group importance and weight magnitude into consideration to evaluate the importance of each weight. LLM-pruner (Ma et al., 2023) evaluates the group importance using gradient-based methods. However, computing gradient for LLMs will introduce a significant amount of memory cost and it is less practical for larger models. The existing unstructured pruning methods (Sun et al., 2024; Frantar & Alistarh, 2023) are more efficient than gradient-based methods. In `Down-Proj` pruning, Wanda prioritizes the weights linked to outliers in intermediate activations. To minimize the impact on these critical intermediate activation outliers, it would be beneficial to also give greater significance to the weights that lead to outlier generation in both `Gate-Proj` and `Up-Proj` projections. Thus, we use the norm of intermediate activations $\|\mathbf{y}\|_2$ as the group importance indicator. The computation of

activation norms is straightforward without introducing more computation and memory overhead than computing gradient. To assign larger importance scores on weights corresponding to more important groups, we evaluate weight importance based on the product of weight magnitude and corresponding group importance score.

More specifically, for the `Gate-Proj` and `Up-Proj`, the importance score attributed to a specific weight $W_{i,j}^{(k)\top}$ is determined as follows:

$$I_{i,j}^{(k)} = \left| W_{i,j}^{(k)\top} \right| \cdot \|\mathbf{y}_i\|_2^\alpha \tag{4}$$

where $k = 1, 2$. We introduce a hyper-parameter group importance strength $\alpha$. For the `Gate-Proj` and `Up-Proj` layers, the intermediate activation $\mathbf{y}$ is itself produced by these projections. We hypothesize that in `Gate-Proj` and `Up-Proj`, large-magnitude intermediate activations may already correspond to a higher proportion of large-magnitude weights. Simply multiplying by $\|\mathbf{y}_j\|_2$ could therefore overly favor certain connections. Using integration strength $\alpha < 1$ still boosts weights associated with large intermediate activations but avoids disproportionately concentrating on them. We empirically find that $\alpha = 0.5$ is a well-balanced point for different models and datasets in our preliminary studies.

For the `Down-Proj` projection pruning, we directly augment group importance into weight magnitude without additional hyper-parameter. For `Down-Proj` matrix, the importance score of weight $W_{i,j}^{(3)\top}$ is determined as follows:

$$I_{i,j}^{(3)} = \left| W_{i,j}^{(3)\top} \right| \cdot \|\mathbf{y}_j\|_2 \tag{5}$$

In the `Down-Proj` layer, the output is literally the elementwise product of the MLP intermediate activation $\mathbf{y}$ and the `Down-Proj` weights. Since this multiplication happens in the forward pass, weighting each connection by $\|\mathbf{y}_j\|_2$ directly reflects its real contribution to the final output. Using Eq. 5 is a natural choice and mirrors Wanda's original formula. Introducing an integration strength $\alpha < 1$ in this layer brings no clear benefit in practice (as shown in Table 12), so we keep $\alpha = 1$ for simplicity. By augmenting intermediate activations into all three weight importance matrices, DaSS inherently assigns greater emphasis to weights corresponding to intermediate activation outliers, thereby facilitating more nuanced structural coordination among the entire MLP module.

**Pruning Granularity.**  Pruning LLMs in finer granularity can improve the performance (Sun et al., 2024). To incorporate intermediate activations into GLU-based MLP pruning, each weight in the same comparison group should correspond to different intermediate activations. Therefore, we choose to use input-balanced pruning for `Gate-Proj` and `Up-Proj` pruning, in which the weights are compared for each input individually. In such pruning granularity, we can augment intermediate activations into `Gate-Proj` and `Up-Proj` pruning metric. In input-balanced pruning, we remove $s\%$ of the weights linked to each input for a predetermined sparsity ratio of $s\%$ based on weight importance scores. DaSS uses output-balanced sparsity for `Down-Proj` pruning, which is the same as Wanda. Within each comparison group, weights are sorted based on their importance scores, and those with the lowest scores are pruned.

**Extension to N:M Sparsity.**  The DaSS pruning design allows for easy adaptation to the N:M sparsity pattern. For `Gate-Proj` and `Up-Proj` the N: M sparsity pattern is formed on an input-balanced basis. This means that for weights connecting to each input neuron, out of every group of M consecutive weights, there are exactly N zeros included. For `Down-Proj` the N: M sparsity pattern is formed on an output-balanced basis.

**Discussion.**  In summary, our DaSS method offers multiple appealing aspects for pruning LLMs:

1. It retains the fundamental simplicity inherent in Wanda pruning method. Without weight updating, it still matches the performance of SparseGPT even at high sparsity as demonstrated in Section 4.4. This demonstrates the consistently effective and efficient capability of the DaSS method in identifying sparse neural networks.

2. Unlike SparseGPT and Wanda that use *input + intermediate* activations for MLP pruning, DaSS only uses *intermediate* activations. By using *intermediate* activations as group importance indicator,

DaSS prunes MLP module in a more comprehensive view that captures the collective importance of all the weights connecting to each intermediate neuron.

3. DaSS effectively explores the balance between unstructured pruning's flexibility and the structural coherence in dependency-based structured pruning.

# 4 Experiments

## 4.1 Settings

**Models.** DaSS's performance is evaluated over open LLMs using GLU variants. SwiGLU is the most widely used GLU-based MLP in the recent LLMs, including the LLaMA2 model family (Touvron et al., 2023), which has models with parameters ranging between 7 billion and 70 billion, and also the Mistral-7B model (Jiang et al., 2023). Among them, LLaMA2-70B and Mistral-7B use grouped-query attention (Ainslie et al., 2023), the MLP module parameter accounts for around 80% of the total model parameters. There are only a few open LLMs that use other variants. For GeGLU, we use Gemma-7B. It is worth noting that the MLP intermediate dimension of Gemma-7B is $8\times$ model dimension, making the MLP module much larger. For ReGLU, we use ReluLLaMA (Team, 2023), which is fine-tuned using ReGLU variant (Shazeer, 2020; Mirzadeh et al., 2023) based on LLaMA2 with small accuracy loss. The model configuration details are in Appendix A.1. We access the public checkpoints of the involved models provided by HuggingFace Transformers (Wolf et al., 2019).

**Baseline Approaches.** We compare the performance with two LLM-specific one-shot pruning approaches, SparseGPT (Frantar & Alistarh, 2023) and Wanda (Sun et al., 2024). We don't consider structured pruning methods like LLM-Pruner (Ma et al., 2023) and Sheared LLaMA (Xia et al., 2023), as they typically require re-training to recover accuracy, and are less practical for large models like LLaMA2-70B. Those baseline methods utilize uniform layerwise sparsity that can be easily converted into hardware-friendly N:M sparsity pattern. We used the same calibration data set as SparseGPT and Wanda in their model pruning processes, consisting of 128 sequences of 2048 tokens each, randomly selected from the first shard of the C4 dataset (Raffel et al., 2020).

**Evaluation.** To comprehensively evaluate the efficacy of our proposed method, two different metrics are utilized to evaluate the performance of the pruned models: (1) perplexity (PPL) of language modeling (2) zero-shot accuracy on 5 commonsense reasoning tasks. Perplexity has been regarded as a consistent and reliable metric for measuring compressed models (Dettmers & Zettlemoyer, 2023; Frantar & Alistarh, 2023), while downstream tasks sometimes have tendency in noisy behavior, but more interpretable. For perplexity evaluation, we use the validation dataset of WikiText2 (Merity et al., 2017). We set the context size for perplexity evaluation as 2048 for all the models. The perplexity evaluation setting of LLaMA2 differs from that used in Sun et al. (2024), resulting in discrepancies between our results and those reported in Sun et al. (2024).[3] For zero-shot commonsense reasoning tasks, we choose five widely used tasks for accuracy evaluation: ARC (Easy and Challenge) (Clark et al., 2018), HellaSwag (Zellers et al., 2019), PiQA (Bisk et al., 2020), and WinoGrande (Sakaguchi et al., 2021), implemented in the `Lm-Evaluation-Harness` (Gao et al., 2021). We evaluate the perplexity of all the aforementioned models. To fully demonstrate the task-wise performance in different sparsity patterns, we report the downstream task performance of the largest LLaMA2-70B model. Notably, LLaMA2-70B utilizes grouped-query attention (Ainslie et al., 2023), and the MLP module accounts for more than 80% of the total parameters, representing the most widely adopted architectural design in modern LLMs.

**Sparsity.** In the less interpretable perplexity evaluation, we only prune the MLP layers. In the task-wise evaluation of the LLaMA2-70B model, both attention and MLP modules are pruned, consistent with prior works to accurately assess the performance gap between pruned and original models. Since DaSS is not applicable to attention module pruning, we employ the Wanda method to prune the attention module,

---

[3]Sun et al. (2024) employs a context size of 4096 tokens and prunes both the MLP and attention modules, whereas our perplexity evaluation only involves pruning the MLP module.

Table 1: WikiText perplexity of pruned LLMs. Here we only prune the MLP module. The (*) in Size indicates a model that uses larger MLP than Vallina Transformer (e.g., grouped query attention).

| MLP Sparsity | Method | PPL (↓) | | | | | | |
| | | Gemma (GeGLU) | ReluLLaMA (ReGLU) | | Mistral (SwiGLU) | LLaMA2 (SwiGLU) | | |
| | | 7B* | 7B | 13B | 7B* | 7B | 13B | 70B* |
|---|---|---|---|---|---|---|---|---|
| Dense | - | 7.00 | 6.15 | 5.50 | 5.25 | 5.47 | 4.88 | 3.32 |
| 4:8 | SparseGPT | **10.86** | 8.44 | 7.45 | 7.36 | 7.33 | 6.29 | 4.66 |
| | Wanda | 14.41 | 9.21 | 7.74 | 7.38 | 7.63 | 6.42 | 4.59 |
| | DaSS | **10.86** | **8.19** | **7.16** | **7.06** | **7.26** | **6.16** | **4.41** |
| 2:4 | SparseGPT | 13.93 | 10.26 | 8.98 | 8.86 | 8.72 | 7.30 | 5.32 |
| | Wanda | 32.50 | 12.68 | 9.87 | 9.24 | 9.55 | 7.68 | 5.35 |
| | DaSS | **13.69** | **9.61** | **8.18** | **8.39** | **8.48** | **6.90** | **4.91** |
| 50% | SparseGPT | 9.64 | **7.22** | **6.39** | 6.20 | **6.38** | 5.60 | 4.06 |
| | Wanda | 9.93 | 7.52 | 6.53 | 6.25 | 6.50 | 5.67 | 4.07 |
| | DaSS | **9.11** | 7.24 | 6.40 | **6.15** | 6.44 | **5.59** | **4.00** |

ensuring overall efficiency remains consistent with Wanda. We apply a uniform sparsity ratio across all the pruned layers and evaluate three sparsity types: unstructured sparsity, and semi-structured sparsities of 4:8 and 2:4.

## 4.2 Language Modeling

We examined all the aforementioned LLMs in terms of perplexity as shown in Table 1. Our method consistently achieves better performance than SparseGPT and Wanda in more constrained and practical N:M sparsity pattern. As indicated by Sun et al. (2024), where weight updates can improve the performance in N:M sparsity pattern, our method shows superior performance even without computationally expensive weight updates. For Mistral-7B, Gamma-7B and LLaMA2-70B models with larger MLP layers, our method also outperforms SparseGPT in unstructured sparsity. Impressively, for the Gemma-7B with much larger MLP layers, DaSS outperforms Wanda in N:M sparsity significantly. DaSS shows consistent effectiveness across SwiGLU, GeGLU, and ReGLU, proving the generalizability of the DaSS method on GLU variants.

## 4.3 Downstream Tasks

Apart from assessing perplexity, we comprehensively evaluate the performance of pruned LLaMA2-70B models in the widely used commonsense reasoning tasks.

In Table 2, we present the performance of different sparse LLaMA2-70B models on downstream tasks with prompting. The results show that our method outperforms SparseGPT and Wanda in most tasks at semi-structured N:M sparsity pattern. The only exception is that SparseGPT outperforms both Wanda and DaSS in Winogrande task. 50% unstructured SparseGPT pruned model even outperforms the dense model. Such results align with the prevailing LLM compression works (Dettmers & Zettlemoyer, 2023; Frantar & Alistarh, 2023), where single task results can be noisy and the mean accuracy provides stable and reliable results. For unstructured sparsity, our method outperforms Wanda sharing the same complexity level.

## 4.4 Performance Analysis

**Sparsity Variation** Table 3 illustrates a comparison of the mean zero-shot task accuracy at varying levels of sparsity for the MLP module of the LLaMA2-70B model. It's evident that DaSS pruning maintains competitive performance which closely matches that of SparseGPT across the entire range of sparsity ratios tested. Notably, DaSS achieves such performance without the need for weight updates, suggesting its effectiveness and efficiency in locating sparse neural networks. On the other hand, output-balanced Wanda pruning shows a significant decline in accuracy as the sparsity ratio increases. This suggests that Wanda pruning may suffer from structural mismatch issues within the MLP layer, which become more pronounced

Table 2: Downstream tasks performance of LLaMA2-70B model in different sparsity pattern.

| Sparsity | Method | CommonSenseQA (↑) | | | | | |
|---|---|---|---|---|---|---|---|
| | | PIQA | HellaSwag | Winogrande | ARC-e | ARC-c | Average |
| Dense | - | 82.15 | 66.05 | 77.98 | 82.55 | 54.35 | 72.62 |
| 4:8 | SparseGPT | 80.52 | 61.00 | **77.03** | 79.85 | 50.60 | 69.80 |
| | Wanda | 80.47 | 61.85 | 75.45 | 80.10 | 50.00 | 69.57 |
| | DaSS | **80.79** | **62.70** | 76.09 | **81.20** | **51.19** | **70.39** |
| 2:4 | SparseGPT | 79.00 | 59.00 | **76.64** | 78.50 | 47.87 | 68.20 |
| | Wanda | 79.22 | 59.25 | 74.66 | 78.90 | 47.01 | 67.81 |
| | DaSS | **79.70** | **60.00** | 74.82 | **79.65** | **49.15** | **68.66** |
| 50% | SparseGPT | **81.50** | 64.20 | **78.45** | **81.90** | **52.73** | **71.76** |
| | Wanda | 81.01 | 64.30 | 77.35 | 80.95 | 52.05 | 71.13 |
| | DaSS | 81.18 | **64.60** | 77.90 | 81.35 | 51.71 | 71.35 |

Table 3: LLaMA2-70B mean zero-shot tasks performance at different MLP sparsity ratios

| MLP Sparsity | 40% | 50% | 60% | 70% | 80% |
|---|---|---|---|---|---|
| SparseGPT | 72.28 | **71.97** | **70.08** | 65.53 | 52.14 |
| Wanda | 72.35 | 71.49 | 69.47 | 62.75 | 40.76 |
| DaSS | **72.51** | 71.89 | 70.05 | **66.38** | **53.00** |

at higher sparsity levels. As a result, the neural network's performance deteriorates, potentially leading to a dysfunctional model at extreme sparsity ratios.

**Robustness to calibration samples.** Ashkboos et al. (2023) observes that the intermediate activations of SwiGLU-based MLP layers exhibit high variance, primarily caused by the Hadamard product of the preceding two outputs. This leads to diminished accuracy in 4-bit activation quantization. In Figure 2, we present how varying the number of sequences sampled for calibration affects the performance of pruning methods. Even though only using intermediate activations, our method demonstrates minimal sensitivity to changes in the number of calibration samples.

**Ablation Study on Hyperparameter $\alpha$** We conducted an additional ablation study to investigate the impact of different values of the group importance strength hyperparameter $\alpha$ on the performance of DaSS. Specifically, we experimented with LLaMA2-7B using $\alpha = \{0.25, 0.5, 0.75, 1.0\}$. The results are presented in Table 4.

The ablation results show that $\alpha = 0.25$ achieves slightly better performance compared to $\alpha = 0.5$ for LLaMA2-7B in terms of WikiText perplexity. However, we note that $\alpha = 1.0$ still outperforms Wanda (PPL: 6.50). We used $\alpha = 0.5$ for all experiments without intentionally tuning this hyperparameter for different models and tasks. In Appendix A.4, we observe that the effect of $\alpha = 0.5$ is more evident for Mistral-7B with grouped-query attention.

## 4.5 Running Time Analysis

**Pruning speed** For DaSS and Wanda, the computational complexity is quantified as $O(d^2)$, while SparseGPT exhibits a higher complexity of $O(d^3)$. We recorded the overall time taken for pruning MLP layers, not including the forward pass process, in accordance with the approach described by Sun et al. (2024). We use a single A6000 48GB GPU to prune the 7B and 13B models, and use 8 A100 40GB GPUs to prune the larger 70B model.

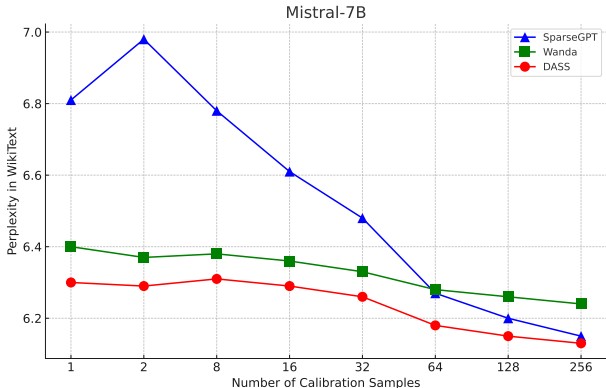

Figure 2: Robustness to calibration samples.

Table 4: WikiText perplexity of 50% MLP sparsity for LLaMA2-7B using different values of $\alpha$.

| $\alpha$ | PPL ($\downarrow$) |
|---|---|
| 1.0 | 6.48 |
| 0.75 | 6.46 |
| 0.5 | 6.44 |
| 0.25 | 6.43 |
| 0.0 | 6.48 |

Table 5: Pruning speed (in seconds) comparison.

| Method | Mistral 7B | LLaMA-2 7B | LLaMA-2 13B | LLaMA-2 70B |
|---|---|---|---|---|
| SparseGPT | 155 | 185 | 206 | 1092.76 |
| Wanda | 0.60 | 0.54 | 0.85 | 14.13 |
| DaSS | 0.81 | 0.81 | 1.02 | 20.70 |

As demonstrated in Table 5, the computational overhead incurred by DaSS is minimal, especially when compared to SparseGPT. Although the processing speed of DaSS is marginally slower than Wanda, this difference arises primarily from the implementation details of *torch.sort()*. In the implementation of *torch.sort()*, sorting along the last dimension is more efficient. Therefore, the observed delay in DaSS is attributed to the sorting operations rather than an inherent inefficiency in our method. Despite this, the substantial improvements in accuracy achieved by DaSS make the additional computational time justifiable and beneficial.

**Inference efficiency** We prune both attention and MLP module to test the inference speed. The speedup achieved by sparse model, as demonstrated in Table 6, highlights a significant reduction in end-to-end decode latency when utilizing LLaMA2-7B on the DeepSparse inference engine (NeuralMagic, 2021). The speed is tested on an Intel Xeon Platinum 8160 CPU with 24 cores. We highlight that the inference speedup is not exclusive to our pruning method but is a result of the inherent power of sparsity in accelerating computation. Critically, different unstructured pruning methods, when achieving the same level of sparsity, tend to exhibit similar inference speedups. This is because the primary factor influencing the speedup is the sparsity level itself, not the specific algorithm used to obtain that sparsity. A detailed comparison of inference speeds for various pruning methods, including SparseGPT, Wanda, and DaSS, is provided in Appendix A.6, further demonstrating this point.

## 5 Related Work

**Pruning LLM.** Neural network pruning in LLM can be broadly categorized into two groups: structured pruning (Ma et al., 2023; Zhang et al., 2023a) and unstructured pruning (Frantar & Alistarh, 2023; Sun et al., 2024). Ma et al. (2023) proposes a dependency detection algorithm to detect and prune non-critical grouped structures followed by LoRA fine-tuning. Although structured pruning can usually achieve better hardware efficiency, the accuracy drops a lot even at a low compression rate. Unstructured pruning can yield a higher compression rate and achieve acceleration on Nvidia's GPUs by employing a hardware-friendly N:M sparsity pattern. SparseGPT (Frantar & Alistarh, 2023) leverages the Hessian inverse for pruning

Table 6: Decode throughput performance for 50% and 70% unstructured sparsity with LLaMA2-7B utilizing DeepSparse NeuralMagic (2021) inference engine.

| Sparsity | Dense | 50% | 70% |
|---|---|---|---|
| **Tokens per seconds** | 3.05 | 5.64 | 8.27 |
| **Speedup** | 1.0× | 1.85× | 2.71× |

and reduces reconstruction error of dense and sparse weights by subsequent weight updates. Boža (2024) introduces an efficient layer-wise weight update strategy based on alternating direction method of multipliers (ADMM). Wanda (Sun et al., 2024) employs an efficient method that augments input activations into weight magnitudes, and matches the performance of SparseGPT at medium sparsity. Our work incorporates dependency information into unstructured pruning, achieving a new pruning paradigm. Notably, both SparseGPT and ADMM rely on weight updates to recover accuracy after pruning, which can introduce significant computational overhead, especially for larger models. In contrast, Wanda and our proposed DaSS method eliminate the need for weight updates. Impressively, DaSS achieves performance comparable to SparseGPT even at high sparsity levels without requiring any weight updates, highlighting its efficiency and effectiveness.

**Inherent Sparsity of Transformer MLP.** Interestingly, sparsity within the MLP activations of trained Transformer-based models occurs innately even without applying explicit regularizations or constraints (Zhang et al., 2022; Li et al., 2023; Dong et al., 2023). Such a phenomenon is prevalent in learned Transformers, including other zero-saturating functions. Liu et al. (2023); Mirzadeh et al. (2023); Zhang et al. (2022) achieve actual LLM inference speedup by only performing computation corresponding to the activating neuron for a given input. They do not actually reduce the model size since they mainly reduce I/O and computation latency in a selective weights loading manner, and thus, these methods are less applicable in large batch-size inference settings. Our work investigates the weight sparsity in MLP module by considering corresponding intermediate activations.

**Outlier-dependent LLM Compression.** Outlier features, defined as features with magnitudes substantially larger than others, are a notable characteristic of LLMs (Dettmers et al., 2022). Despite making up only a small fraction of all feature dimensions, these outliers play a critical role in attention and predictive performance. Such observation has motivated the development of LLM-specific quantization methods (Dettmers et al., 2022; Xiao et al., 2023; Lin et al., 2024; Ashkboos et al., 2023) to handle outliers more effectively. Wanda (Sun et al., 2024) and OWL (Yin et al., 2024) broaden these insights, revealing that outlier features are significant in deciding weight importance when pruning LLMs. Our method diverges from conventional wisdom by demonstrating that, in the context of GLU-based MLP input projections, the inportance of input activation outliers is not as pronounced as previously assumed, prompting a reevaluation of their role in LLM pruning strategies.

## 6 Conclusion

We propose the Dependency-aware Semi-structured Sparsity (DaSS) method which effectively addresses the challenges of pruning GLU-based MLP modules LLMs. DaSS strikes a unique balance between the adaptability of unstructured pruning and the orderliness of structured pruning. By leveraging the MLP intermediate activation norms as the group importance indicator, we develop a novel pruning metric that assesses weight importance in a more structurally consistent manner. Empirical evaluations on the Mistral, Gemma and LLaMA2 model families show that DaSS surpasses state-of-the-art pruning methods such as SparseGPT and Wanda in achieving hardware-friendly N:M sparsity patterns. We hope our work inspires further research and development in the compression of GLU-based LLMs.

**Limitations.** For `Gate-Proj` and `Up-Proj`, the pruning directions differ from those used in Wanda and SparseGPT. In the CUTLASS library, the pruning direction for 2:4 sparsity is fixed and cannot be modified. Conversely, the cuSPARSELt library supports transposition fusion, offering more flexibility. However,

2:4 sparsity is still a prototype feature in PyTorch, which leads to several limitations, including incomplete functionality and unresolved bugs. Furthermore, the usage of transposition fusion of cuSPARSELt in PyTorch has not been thoroughly documented. We anticipate that PyTorch will address these issues in future updates, improving the usability and robustness of 2:4 sparsity features.

In Table 2, models with 2:4 sparsity exhibit noticeable accuracy degradation compared to dense models. However, this issue can be alleviated by adopting more flexible sparsity patterns such as 4:8 sparsity. Although Kurtic et al. (2023) has reported inference speed-ups using even more flexible patterns like 32:64 sparsity, the lack of support for these advanced semi-structured sparsity patterns in PyTorch underscores the need for further efforts from the research community to enable their practical implementation.

A limitation of our current work is that it primarily targets the reduction of model weight size and computational cost. In long-context inference, where the Key-Value (KV) cache size often becomes the dominant memory bottleneck, the benefits of model weight pruning alone are less pronounced. Recent studies have increasingly focused on KV cache compression (Zhang et al., 2023b; Xiao et al., 2024; Guo et al., 2024) to tackle this challenge. While outside the scope of this paper, we recognize the exciting potential of combining model weight pruning with these emerging KV cache reduction strategies. Future work exploring this synergy could lead to more comprehensive solutions for efficient long-context inference, addressing both model weight and KV cache related memory challenges.

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

# A Appendix

## A.1 Model Configurations

Table 7 is the configurations of the models used in the paper. We did not use LLaMA2-34B as it was not released. ReluLLaMA uses the same configuration as LLaMA2, with the only difference being the activation function. The links to the ReluLLaMA models are provided below:

- ReluLLaMA-7B: `https://huggingface.co/SparseLLM/ReluLLaMA-7B`

- ReluLLaMA-13B: `https://huggingface.co/SparseLLM/ReluLLaMA-13B`

Table 7: Model configurations of Llama2, Gemma and Mistral models.

| Model | Param | Layers | Hidden | Intermediate | Query Heads | KV Head |
|-------|-------|--------|--------|--------------|-------------|---------|
| LlaMa2-7B | 7B | 32 | 4096 | 11008 | 32 | 32 |
| LlaMa2-13B | 13B | 40 | 5120 | 13824 | 40 | 40 |
| LlaMa2-70B | 70B | 80 | 8192 | 28672 | 64 | 8 |
| Mistral-7B | 7B | 32 | 4096 | 14336 | 32 | 8 |
| Gemma-7B | 7B | 28 | 3072 | 24576 | 16 | 16 |

Table 8: The Wikitext perplexity results of LLaMA2-7B with 50% sparsity and 4 bit quantization

| Method | SaprseGPT | Wanda | DaSS |
|--------|-----------|-------|------|
| 50% sparsity | 6.38 | 6.50 | 6.44 |
| w/ 4-bit AWQ | 6.51 | 6.63 | 6.57 |

## A.2 Combination with Quantization

Pruning and quantization, traditionally seen as distinct model compression techniques, are not mutually exclusive and can be effectively integrated to enhance overall efficiency (Han et al., 2015a; Frantar & Alistarh, 2023; Agarwalla et al., 2024). Here we test the combination of DaSS with 4-bit AWQ (Lin et al., 2024) for compressing LlaMA2-7B. As shown in Table 8, the performance of all the pruned models just drops slightly after 4-bit quantization. DaSS still outperforms wanda after 4-bit quantization.

## A.3 MMLU Results

Recent work (Gromov et al., 2024) indicates the unusual behavior of LLMs in performing Massive Multitask Language Understanding (MMLU) (Hendrycks et al., 2021) tasks. We use `Chain-of-Thought Hub` (Fu et al., 2023) which is based on the official implementation of MMLU (Hendrycks et al., 2021). The MMLU encompasses 57 tasks, spanning from STEM, Humanities, Social Sciences, among others and we report the mean accuracy of 57 tasks. We test the performance of pruned LlaMA2-70B model in MMLU task. From Table 9, we observe that the improvement of DaSS over Wanda becomes more pronounced in the challenging MMLU task, where it achieves an increase in accuracy of 1.17 and 1.12 for the 4:8 and 2:4 sparsity patterns, respectively. In accordance with the observations with Jaiswal et al. (2023), we see a considerable performance degradation in knowledge-intensive MMLU task, in which only unstructured 50% sparsity models outperform the dense LLaMA2-34B model. As GPU plays a more important role in larger model inference, it is essential to improve the performance of pruned models in hardware-friendly N:M sparsity. Uniform reduction of the overall sparsity level is not feasible for N:M sparsity, Frantar & Alistarh (2023) suggests a specific subset of layers can be chosen for full N:M sparsification. Here we skip pruning 1/4 consecutive layers (20 layers) and result in a final 37.5% sparsity ratio. We use the first 10 tasks of MMLU to study pruning sensitivity. We divide the model into 4 consecutive parts and study skipping pruning each part. As shown in Table 10, *the earlier layers are more sensitive than the later ones in knowledge-intensive tasks*, which is contradictory to the findings in Frantar & Alistarh (2023) using perplexity, and align with the findings of Gromov et al. (2024).

We continue to search for the better skipping layers with the start layer index range in [0,20]. We found that starting skipping from layer 10 can achieve the best performance in the subset. Then we test its results in the full MMLU tasks. As shown in Table 2, skipping sensitive 1/4 layers can significantly improve the performance of pruned models, especially for N:M sparsity. We can achieve sparse models that perform better than LLaMA2-34B. Although partial N:M sparsity models have more parameters than smaller dense models, training many smaller dense models like LLaMA2-34B is still computationally expensive. Efficient post-training pruning enables us to easily adjust the accuracy-efficiency trade-off in real-world applications.

Recent work (Gromov et al., 2024) propose an effective method to remove around half of the layers in LlaMA2-70B with small accuracy loss for MMLU tasks, suggesting the unusual behavior of LLMs in performing

Table 9: MMLU scores for different sparsity methods and levels

| Sparsity | Methods | MMLU (5 shot) |
|---|---|---|
| Dense | - | 69.10 |
| 4:8 | SparseGPT | 60.24 |
|  | Wanda | 59.69 |
|  | DaSS | **60.86** |
|  | DaSS+skip 1/4 | 65.82 |
| 2:4 | SparseGPT | 56.99 |
|  | Wanda | 56.41 |
|  | DaSS | **57.53** |
|  | DaSS+skip 1/4 | 64.36 |
| 50% | SparseGPT | **64.52** |
|  | Wanda | 62.72 |
|  | DaSS | 63.27 |
|  | DaSS+skip 1/4 | 66.81 |

Table 10: MMLU subset accuracy after skipping pruning 20 layers at various start indices in 4:8 sparsity.

| Start Index | 0 | 10 | 20 | 40 | 60 | Dense |
|---|---|---|---|---|---|---|
| Acc (%) | 60.12 | 61.72 | 59.75 | 56.54 | 56.83 | 64.86 |

MMLU tasks. Their work is focused more on task-specific pruning, while our work emphasizes preserving the general ability of LLMs to perform various tasks. Our weight pruning method is fundamentally orthogonal to layer pruning in principle.

### A.4 $\alpha$ ablation study using Mistral

Due to the use of grouped-query attention (Ainslie et al., 2023) in Mistral-7B, the MLP module constitutes a relatively larger portion of the model's parameters. Therefore, we conducted a study to investigate the influence of the integration strength, $\alpha$, specifically on the `Gate-Proj` and `Up-Proj` components within Mistral-7B's MLP module. Table 11 presents the results of this study under both 50% unstructured and 2:4 sparsity settings. As the table shows, the impact of using $\alpha = 0.5$ is more pronounced in this context.

To evaluate the impact of the integration strength parameter, $\alpha$, on the `Down-Proj` component, we conducted experiments using Mistral-7B with 50% unstructured sparsity, as shown in Table 12. While maintaining $\alpha$ at 0.5 for both `Gate-Proj` and `Up-Proj`, we varied $\alpha$ in `Down-Proj`. The nearly identical perplexity scores obtained with $\alpha = 1.0$ and $\alpha = 0.5$ suggest that using $\alpha$ in `Down-Proj` provides minimal to no benefit. Therefore, in the interest of simplicity, we did not employ an $\alpha$ parameter for `Down-Proj` in our main experiments.

### A.5 LLaMA3.1 results

To further validate the effectiveness of our proposed DaSS method, we conducted additional experiments using the recently released LLaMA-3.1-8B model. LLaMA-3.1 shares the same architectural components as LLaMA2-70B, including grouped-query attention and SwiGLU, making it highly compatible with our approach. The results demonstrate the consistent robustness of DaSS , as shown in the table 13.

The LLaMA-3.1-405B model, being the most appealing variant for model compression research, presents an exciting avenue for future exploration. However, due to computational resource constraints, we were unable to conduct experiments on this larger model at this time. We hope to address this limitation in subsequent work when resources permit.

Table 11: WikiText perplexity of 50% unstructured and 2:4 sparsity for Mistral-7B using different $\alpha$

| $\alpha$ | 50% | 2:4 |
|---|---|---|
| 1.0 | 6.27 | 8.84 |
| 0.75 | 6.20 | 8.61 |
| 0.5 | **6.15** | 8.39 |
| 0.25 | **6.15** | **8.33** |
| 0.0 | 6.32 | 8.61 |

Table 12: WikiText perplexity of 50% unstructured sparsity for Mistral-7B using different $\alpha$ in `Down-Proj`

| $\alpha$ | 50% |
|---|---|
| 1.0 | 6.15 |
| 0.75 | **6.14** |
| 0.5 | 6.15 |
| 0.25 | 6.21 |
| 0.0 | 6.37 |

## A.6   Inference Speed Comparison

We conducted an evaluation of inference speed for different pruning methods under 50% and 70% sparsity on CPU. The results are presented in Table 14.

As illustrated in Table 14, the inference speed differences between the sparse models (SparseGPT, Wanda, and DaSS) at the same sparsity level are negligible. This observation aligns with the typical behavior of unstructured pruning methods, where the primary determinant of inference speed is the sparsity level itself, rather than the specific pruning algorithm employed.

It is important to reiterate that, consistent with other unstructured pruning research, our main contribution lies in enhancing task performance (perplexity and accuracy) for a given sparsity level, while maintaining a pruning efficiency comparable to that of Wanda. The results demonstrate that although DaSS does not provide significant gains in inference speed compared to other unstructured methods at the same sparsity, it achieves superior accuracy, making it a more effective pruning approach overall.

Table 13: WikiText perplexity of pruned LlaMA-3.1-8B models

| Method | MLP Sparsity | PPL ($\downarrow$) |
|---|---|---|
| Dense | - | 6.23 |
| SparseGPT | 4:8 | 10.01 |
| Wanda | 4:8 | 10.40 |
| DaSS | 4:8 | **9.95** |
| SparseGPT | 2:4 | 12.40 |
| Wanda | 2:4 | 13.94 |
| DaSS | 2:4 | **12.18** |
| SparseGPT | 50% | 8.41 |
| Wanda | 50% | 8.41 |
| DaSS | 50% | **8.35** |

Table 14: Inference speed (tokens/second) for different sparse models at 50% and 70% sparsity on CPU.

| Model | 50% Sparsity | 70% Sparsity |
|---|---|---|
| SparseGPT | 5.64 | 8.23 |
| Wanda | 5.63 | 8.28 |
| DaSS | 5.64 | 8.27 |
| Dense | 3.05 | |

