# OpenReview forum: "Dependency-Aware Semi-Structured Sparsity of GLU Variants in Large Language Models"
_TMLR — Accepted by TMLR_

### Review · Reviewer_GruC · 2024-10-21

**Summary Of Contributions:**

Authors propose to change the Wanda pruning criterion (`weight * <activation norm>`) for MLP blocks in LLMs.
The proposed change is to use `weight * <norm of intermediate MLP activation>`.
Pruning of attention blocks uses Wanda-based criteria.

**Audience:**

No

**Broader Impact Concerns:**

None.

**Claims And Evidence:**

No

**Requested Changes:**

- Fix the numbers, so they match Wanda paper, or explain the differences.
- Do 2:4 sparsity correctly, so it is usable in actual hardware.
- Provide actual code.

**Strengths And Weaknesses:**

## Strengths

Results are better than Wanda pruning.

## Weaknesses

### Numbers in Table 1 are strange.
For Llama2-7B and 2:4 pruning Wanda paper reports Wanda perplexity of 11.02, here we have 9.55.
For unstructured 50% pruning, Wanda's numbers also differ, although slightly less (6.42 vs 6.50).
Wanda is a very simple method; evaluation numbers should not differ by more than 1%.

I think the problem is hidden here:
```
For Gate-Proj and Up-Proj the N: M sparsity pattern is formed on an input-balanced basis. This
means that for weights connecting to each input neuron, out of every group of M consecutive weights, there
are exactly N zeros included. For Down-Proj the N: M sparsity pattern is formed on an output-balanced
basis.
```
Authors use N:M sparsity sometimes in a row-wise and sometimes in a column-wise pattern (since there is no supplementary code, it is hard to check). But Wanda and others always select two inputs out of four. This is because hardware mainly supports picking two input values out of four during matrix multiplication (see https://arxiv.org/pdf/2104.08378 figure 2).
So, I believe, that the resulting pruned model **cannot be run on modern HW**.

### The proposed change seems very simplistic

The only change compared to Wanda is that we calculate importance scores by using the intermediate MLP activation norm (compared to Wanda always using the input activation norm). This is a very simplistic change, and I believe it does not warrant a paper. I also believe that for MLP blocks, one could design more theoretically motivated pruning criteria (e.g., one can probably use Down-proj to provide weighting for pruning of Up/Gate-proj).
Also, it would be much more interesting if the authors also figured out a solution for attention blocks.

### Misc

Nitpick: "Fast and Effective Weight Update for Pruned Large Language Models" (published in TMLR https://openreview.net/forum?id=1hcpXd9Jir) gives better results than SparseGPT in similar time.

---

> ### Author Response · Authors · 2024-11-18
> **Author Response to Reviewer GruC**
>
> First, we would like to thank you for taking the time to review our work. Next, we would like to address all the weaknesses pointed by you point-by-point below:
>
> **W1:** Numbers in Table 1 are strange.
>
> Thank you for raising this great question. The perplexity evaluation method of our work differs from Wanda paper in two aspects:
> - (1) Context size: we set the context size as 2048 in all the PPL evaluations. However, in the PPL evaluation of Wanda paper, they set the context length as maximum length of the model. For Llama2, the context size for PPL evaluation is 4096. The author also discussed the PPL differences caused by inconsistent context size https://github.com/locuslab/wanda/issues/47#issuecomment-2099031502 . The maximum context size of different models is not the same, the evaluation method of Wanda paper will lead to weird results for some models with long context. The most widely adopted PPL evaluation method in LLM compression area is fixing the context size as 2048 for all the models. In Table 1, the dense Llama2 7/13/70B model PPL is 5.47/4.88/3.32, which exactly matches other papers such as [1][2] that also use context size 2048.
> We will clarify this point clearly in our paper.
> - (2) Skipping pruning attention module: Different from previous works that prune both attention and MLP, in the perplexity evaluation, we only prune the MLP module to prevent the influence of the attention module. Since we used many models in PPL evaluation, some models use multi-head attention, while others use grouped-query attention. In our table, we denoted "MLP Sparsity", not "Sparsity" in Wanda paper. In the downstream tasks performance of Llama2-70B, we prune both attention and MLP.
>
> [1] AWQ: Activation-aware Weight Quantization for LLM Compression and Acceleration. MLSys 2024
> [2] Beyond Size: How Gradients Shape Pruning Decisions in Large Language Models. Arxiv 2023
>
> **W2:** Do 2:4 sparsity correctly.
>
> Thanks for your knowledgeable comment. Yes, for the gate_proj and up_proj, the pruning directions are different from Wanda and SparseGPT.  In CUTLASS library, the pruning direction of 2:4 sparsity cannot be changed. However, in cuSPARSELt library, the transposition operation is supported. https://github.com/pytorch/pytorch/blob/aaefa484417a84a2f5cfcfe4d5041b596c074b4c/torch/sparse/semi_structured.py#L532 .   For gate_proj and up_proj, we can create row-wise 2:4 sparsity and use the transpose operation to make it become column-wise. We will clarify this point in our paper.
>
> **W3:** The proposed change seems very simplistic
>
> We acknowledge (in fact, happily) the conceptual simplicity of our work and **firmly believe that such simplicity is a strength, rather than a weakness**.  Simplicity and effectiveness are core principles of impactful research in deep learning, enabling methods to be robust, reproducible, and more easily deployable in real-world settings.
> We would like to reference the ACL ARR Reviewer Guidelines (https://aclrollingreview.org/reviewerguidelines ), which explain why simplicity should not be considered a weakness in research :
> > Heuristic 7: This method is too simple.
> > Why this is problematic: The goal is to solve the problem, not to solve it in a complex way. Simpler solutions are in fact preferable, as they are less brittle and easier to deploy in real-world settings.
>
> We also kindly draw your attention to Reviewer 5FbZ’s assessment, which explicitly highlighted *“The proposed method is simple and effective”* as a strength of our work.
>
> **attention module:** it would be much more interesting if the authors also figured out a solution for attention blocks.
>
> Thank you for your insightful comment. We appreciate the suggestion to explore solutions for attention blocks. However, with the growing interest in long-context inference, the memory bottleneck of the attention module primarily arises from the KV cache rather than the model weights. To address this, recent LLMs have widely adopted grouped-query attention (GQA), which significantly reduces the memory cost of the KV cache. Consequently, the parameter size of the attention module has become much smaller compared to the MLP modules.
>
> Moreover, an emerging and promising research direction focuses on reducing the KV cache memory footprint through token pruning within the attention module.[3][4]
>
> [3]H2O: Heavy-Hitter Oracle for Efficient Generative Inference of Large Language Models. NeurIPS 2023
> [4]Efficient Streaming Language Models with Attention Sinks. ICLR 2024
>
> **Nitpick:** "Fast and Effective Weight Update for Pruned Large Language Models"
>
> Thank you for introducing this great work for us, we will discuss it in the related work section.
>
> **Provide actual code.**
>
> Thank you for your suggestions, we have uploaded it.
>
> We sincerely hope our responses answer your questions, and we again take this opportunity to thank you for reviewing our work. We would be very happy to respond to any further questions you might have.

---

> > ### Comment · Reviewer_GruC · 2024-11-18
> > **Transposed matmul does not work in Pytorch**
> >
> > Thank you for your responses.
> >
> > W1: Thank you for the explanation; now everything makes sense.
> >
> > W3: If you take this argument "ad absurdum", then you can, in some cases, split one paper into multiple ones. In this case, I would rather see a paper along the lines of "How to prune and update MLP layers", but this is your decision and I am fine with it for now.
> >
> > W2: The transposition of the sparse matrix is **not supported** in Pytorch. I run a simplistic test (with Pytorch 2.5):
> > ```
> > import torch
> > from torch.sparse import to_sparse_semi_structured
> > A = torch.Tensor([0, 0, 1, 1]).tile((128, 32)).half().cuda()
> > A_sparse = to_sparse_semi_structured(A)
> >
> > inp = torch.randn(256,128).cuda().half()
> > # End of preparation
> > torch.mm(inp, A_sparse.t()) # this works
> > torch.mm(inp, A_sparse) # this does not
> > torch.mm(A_sparse, inp.T) # this works
> > torch.mm(A_sparse.t(), inp.T) # this does not
> > ```
> >
> > As you can see, only cases where 2:4 matmul works is when you have sparsity with regards to input features (you select two inputs out of four). If you have sparsity with regards to outputs (to write to two features out of four) it does not work.
> >
> > However, it is possible that Pytorch does not have this support. If you have a working CUSPARSELt example, I would like to see it.

---

> > > ### Author Response · Authors · 2024-11-22
> > > **Updated Limitations**
> > >
> > > Thank you for your response, especially for providing detailed examples with PyTorch. We are glad to have addressed some of your concerns.
> > >
> > > Regarding the issue of 2:4 sparsity and transpose functionality, PyTorch currently lists cuSPARSELt’s transpose fusion as a way to specify row/column layouts of the result matrix. However, as 2:4 sparsity remains a prototype feature in PyTorch, its documentation and support are incomplete, which leads to functionality gaps and unresolved issues.
> > >
> > > We have updated the Limitations section of our manuscript to clearly indicate these challenges and provide context for the current state of 2:4 sparsity in PyTorch. We hope this addition clarifies the constraints and sets realistic expectations for this feature. Thank you again for pointing this out, and we appreciate your valuable feedback.

---

### Review · Reviewer_gEbX · 2024-11-08

**Summary Of Contributions:**

1. This paper proposes Dependency-aware Semi-structured Sparsity (DaSS), a pruning method for GLU-based LLMs that balances adaptability and structural consistency to address memory and latency challenges in large models. By considering both weight magnitude and activation norms, DaSS achieves efficient N:M sparsity patterns.

2. Empirical tests on LLaMA2, Mistral, and Gemma models show that DaSS outperforms SparseGPT and Wanda in hardware efficiency while maintaining computational performance.

**Audience:**

Yes

**Claims And Evidence:**

Yes

**Requested Changes:**

- It is recommended that the authors evaluate the memory efficiency of the compressed model on actual GPU hardware to validate its practical utility.

- Testing the inference speed of the compressed model on long-text tasks, such as with LongBench, would strengthen the analysis of real-world performance.

- Conducting experiments on newer models like LLaMA-3 could provide further insights into the applicability of the proposed method.

- Evaluating the model on complex reasoning datasets, such as GSM8K, could demonstrate its robustness in handling advanced reasoning tasks.

**Strengths And Weaknesses:**

### Strengths
- The paper is well written and clear.

- Introduces Dependency-aware Semi-structured Sparsity (DaSS), balancing adaptability and structural consistency in pruning GLU-based LLMs.

- Achieves hardware-friendly N: M sparsity patterns, outperforming existing methods like SparseGPT and Wanda in efficiency.

- Maintains model performance even under high sparsity, surpassing SparseGPT without requiring weight updates.

- Demonstrates effectiveness across various GLU variants (SwiGLU, GeGLU, ReGLU) and model families (LLaMA2, Gemma, Mistral), showing adaptability and robustness in diverse architectures.

### Weaknesses

- The authors did not evaluate the memory efficiency of the compressed model on actual GPU hardware.
- The authors did not test the inference speed of the compressed model on long-text tasks (e.g., using LongBench).
- The authors did not conduct experiments on newer models such as LLaMA-3.
- The authors did not evaluate the model on complex reasoning datasets, such as GSM8K.

---

> ### Author Response · Authors · 2024-11-18
> **Author Response to Reviewer gEbX**
>
> We first like to thank you for your time to review our work.
> Next, we would like to address all the weaknesses pointed by you point-by-point below:
>
> **W1:** The authors did not evaluate the memory efficiency of the compressed model on actual GPU hardware.
>
> As introduced in PyTorch documentation (https://pytorch.org/docs/stable/sparse.html#sparse-semi-structured-tensors), the memory compression ratio is 56.25% for torch.float16 or torch.bfloat16, and 62.5% for torch.int8 in 2:4 sparsity pattern. To comprehensively benchmark inference efficiency in GPU, we should use an inference engine. The most widely used LLM inference engine vLLM (https://docs.vllm.ai/en/latest) is working for the integration of 2:4 sparsity. However, at this moment, its documentation has not yet provided guidelines on how to utilize 2:4 sparsity. We anticipate future updates will address this and enable a more thorough evaluation of GPUs. We have updated the Limitations section of our manuscript to clearly indicate these challenges and provide context for the current state of software support for 2:4 sparsity.
>
> **W2:** The authors did not test the inference speed of the compressed model on long-text tasks (e.g., using LongBench).
>
> Thank you for raising this point. During generation, the key and value tensors of previously generated tokens, known as the KV cache, must be preserved in memory for attention computation. The memory cost of the KV cache scales linearly with both the batch size and sequence length. When the context length becomes very large, the memory cost is primarily driven by the KV cache rather than the model weights.
>
> Our work focuses on model weight compression, not KV cache compression. While our method effectively reduces the memory and computational cost associated with the model weights, it does not directly address the challenges of KV cache memory in long-context inference.
>
> Recent studies [1][2][3][4] have explored token pruning to reduce KV cache size, thereby improving the efficiency of long-context inference. For this purpose, LongBench has been widely adopted in KV cache compression research [3][4]. We believe integrating these techniques could complement our work in addressing long-context scenarios and is an exciting direction for future research.
>
> [1]H2O: Heavy-Hitter Oracle for Efficient Generative Inference of Large Language Models. NeurIPS 2023
> [2]Efficient Streaming Language Models with Attention Sinks. ICLR 2024
> [3]SnapKV: LLM Knows What You are Looking for Before Generation. NeurIPS 2024
> [4]Attention Score is not All You Need for Token Importance Indicator in KV Cache Reduction: Value Also Matters. EMNLP 2024
>
> **W3:** The authors did not conduct experiments on newer models such as LLaMA-3.
>
> Thank you for your feedback. We have already conducted extensive experiments on a wide range of open LLMs with diverse architectures, demonstrating the consistent effectiveness of our method across these models. Regarding LLaMA-3, it shares the same architectural components as LLaMA2-70B, including grouped-query attention and SwiGLU, making it highly compatible with our approach.
>
> In response to the reviewer’s interest, we conducted additional experiments using the recently released LLaMA-3.1-8B. The results further confirm the robustness of DaSS in achieving effective model compression, as shown in the table below.
>
> *Table: WikiText perplexity of pruned LlaMA-3.1-8B models*
> | Method    | MLP Sparsity | PPL (↓) |
> |-----------|--------------|---------|
> | Dense     | -            |6.23    |
> |
> | SparseGPT | 4:8          | 10.01  |
> | Wanda     | 4:8          | 10.40   |
> | DaSS      | 4:8          | **9.95**   |
> |
> | SparseGPT | 2:4          | 12.40   |
> | Wanda     | 2:4          | 13.94   |
> | DaSS      | 2:4          | **12.18**   |
> |
> | SparseGPT | 50%          | 8.41   |
> | Wanda     | 50%          | 8.41   |
> | DaSS      | 50%          | **8.35**    |
>
> The most attractive variant of the LLaMA-3.1 series for model compression research is LLaMA-3.1-405B. However, due to computational resource constraints, we were unable to include experiments with LLaMA-3.1-405B at this time. We hope to address this limitation in future work when resources permit.
>
> **W4:** The authors did not evaluate the model on complex reasoning datasets, such as GSM8K.
>
> In Appendix A.3, we evaluated our method on MMLU[5], a comprehensive benchmark covering 57 tasks across diverse domains, including STEM, Humanities, and Social Sciences. While GSM8K focuses exclusively on mathematical reasoning, MMLU includes several math-related subjects alongside a broader spectrum of tasks, making it a more extensive evaluation dataset.
>
> [5] Measuring Massive Multitask Language Understanding. ICLR 2021
>
> We sincerely hope our responses answer your questions, and we again take this opportunity to thank you for reviewing our work. We would be very happy to respond to any further questions you might have.

---

> > ### Comment · Reviewer_gEbX · 2024-11-24
> > **Response to authors**
> >
> > Dear Authors,
> >
> > Thank you for the author's response, which partially addressed my concerns. I am pleased with the progress. However, I am still eager to see the performance of the method proposed in the paper on generative datasets such as GSM8K and LongBench. This would provide insights into how the compressed model performs on generative and long-form generative tasks. I am particularly interested in this aspect. If the authors have conducted experiments in this direction, regardless of the results, it would contribute meaningfully to the analysis of the compressed model, as the primary task of LLMs is generative reasoning. **If the authors include experiments in this area, I promise to recommend acceptance of this paper**.
> >
> > Bests,
> >
> > Reviewer gEbX

---

> > > ### Author Response · Authors · 2024-11-27
> > > **Clarification of your concerns (2/2)**
> > >
> > > **Reference**:
> > >
> > > [1] Do Emergent Abilities Exist in Quantized Large Language Models: An Empirical Study. LREC-COLING 2024
> > >
> > > [2] The case for 4-bit precision: k-bit Inference Scaling Laws. ICML 2023
> > >
> > > [3]Pruner-Zero: Evolving Symbolic Pruning Metric from scratch for Large Language Models. ICML 2024
> > >
> > > [4]Outlier Weighed Layerwise Sparsity (OWL): A Missing Secret Sauce for Pruning LLMs to High Sparsity. ICML 2024
> > >
> > > [5] QuIP#: Even Better LLM Quantization with Hadamard Incoherence and Lattice Codebooks. ICML 2024
> > >
> > > [6] AlphaPruning: Using Heavy-Tailed Self Regularization Theory for Improved Layer-wise Pruning of Large Language Models. NeurIPS 2024
> > >
> > > [7] QBB: Quantization with Binary Bases for LLMs. NeuIPS 2024
> > >
> > > [8]DB-LLM: Accurate Dual-Binarization for Efficient LLMs. ACL 2024 findings
> > >
> > > [9] VPTQ: Extreme Low-bit Vector Post-Training Quantization for Large Language Models. EMNLP 2024
> > >
> > > [10] Prefixing Attention Sinks can Mitigate Activation Outliers for Large Language Model Quantization. EMNLP 2024
> > >
> > > [11]QUIK: Towards End-to-end 4-Bit Inference on Generative Large Language Models. EMNLP 2024

---

> > > > ### Comment · Reviewer_gEbX · 2024-12-20
> > > >
> > > > Thanks for the efforts made by the authors during the rebuttal process. **I agree to accept this paper**. For future work, I suggest the authors include additional generated experiments to make the paper more solid.

---

> ### Author Response · Authors · 2024-11-27
> **Clarification of your concerns (1/2)**
>
> Dear Reviewer gEbX,
>
> Thank you for your continued feedback. We are pleased that we have been able to address some of your concerns, and we would like to respond to your remaining points as follows:
>
> **GSM8K Evaluation**:
>
> We apologize for not fully clarifying this aspect in our previous response. In the field of LLM compression, the performance differences between models are often quite subtle, requiring an evaluation metric capable of capturing these small variations in accuracy. [2] discussed why using perplexity and mean accuracy for evaluating compressed LLMs. Although GSM8K is commonly used for LLM evaluation, it has not been widely adopted in recent LLM compression research due to its limitations in capturing these fine differences. Specifically, when two models have very similar performance, GSM8K results can be noisy, often failing to reflect subtle differences effectively. For instance, [1] used GSM8K to evaluate various quantized models, and Table 1 shows several instances where lower-bit models outperform higher-bit models, which contradicts general expectations that higher-bit models should consistently outperform lower-bit models. This suggests that **GSM8K's noisy results are less reliable for comparing different compressed LLMs**. There are two main reasons that make GSM8K less reliable:
>
> 1. **Limited Number of Test Instances**: GSM8K contains only 1.3k test instances, which makes it less reliable for detecting subtle differences between models. In comparison, MMLU has 14k test instances, while zero-shot tasks like Hellaswag include 10k instances, and we aggregate results from five zero-shot tasks to obtain a mean accuracy, providing a more robust evaluation.
>
> 2. **Challenges in Automatic Evaluation of Generation**: GSM8K often requires chain-of-thought generating, where models generate a reasoning process along with the final answer. This complexity can lead to inconsistencies in how models present their answers, making automatic scoring challenging and potentially introducing noise into the results.
>
> Due to these factors, recent LLM compression studies such as [3]–[11] have generally not included GSM8K in their evaluations. We aimed to align our evaluation with current best practices to ensure consistency and comparability with existing work.
>
> **LongBench Evaluation**:
>
> Based on your suggestion, we investigated the use of LongBench for evaluation. However, we found that it is not supported in the widely used LLM evaluation framework, lm-evaluation-harness ([https://github.com/EleutherAI/lm-evaluation-harness/tree/main](https://github.com/EleutherAI/lm-evaluation-harness/tree/main)). LongBench is primarily used for evaluating KV cache compression, and very few model compression studies have utilized it for evaluating compressed LLMs. Consequently, there is no readily available code for evaluating pruned or compressed LLMs on LongBench, and implementing such an evaluation framework from scratch would require a significant amount of time and effort.
>
> In addition, LongBench comprises 16 English tasks, and the nature of long-context generative inference makes the evaluation process extremely time-consuming. These factors together imply that **conducting experiments on LongBench would demand substantial computational resources and time**, which is not feasible given the current constraints. Therefore, we believe that focusing on more widely adopted evaluation metrics that provide reliable and consistent insights is the most effective approach for our study.
>
> **Summary**:
>
> We acknowledge that our evaluation method is not perfect, and it is nearly impossible to comprehensively evaluate the capabilities of LLMs in a conference-length paper. Efficiently and accurately measuring the performance of compressed LLMs is an important and ongoing research topic. The primary contribution of our work lies in proposing a new model pruning method and using well-established metrics to verify its effectiveness. Investigating new methods to evaluate compressed LLMs does not align with the primary contribution of our work. We have used a sufficient number of metrics (perplexity, five zero-shot tasks, MMLU) to verify the effectiveness of our approach. Compared to recent LLM compression works [3]-[11] published in ICML, NeurIPS, ACL, and EMNLP, our evaluation metrics are on par with those works. Therefore, we believe that **imposing a requirement for extensive additional evaluations is not a fair acceptance criterion for our work**.
>
> We hope that our response addresses your concerns.
>
> Best regards,
> The Authors

---

### Review · Reviewer_5FbZ · 2024-11-13

**Summary Of Contributions:**

This paper proposes to take into account the dependency structure of weights for semi-structured model pruning by using the activation norm after the gate proj and up proj linear layers to define the group importance that is multiplied by the absolute value of the weight itself to define the importance of each weight in the network. This importance metric is then used to prune the network. The proposed method (DaSS) achieves superior performance than Wanda and also comparable / superior performance than SparseGPT which updates the remaining weights to compensate for the removal, hence significantly increasing the time required to prune the model.

**Audience:**

Yes

**Claims And Evidence:**

Yes

**Requested Changes:**

- Justify the selection of 0.5 which only provide marginal improvement over the default choice of 1 in table 4
- Update table 6 and the corresponding text to make the run time comparison clear as it is very unclear at this point
- Include Wanda for attention and DaSS for MLP in benchmark results, which was used to report speedup in table 6

**Strengths And Weaknesses:**

## Strengths
- The paper is well written and easy to follow
- The proposed method is simple and effective
- Achieves comparable or superior performance than SparseGPT which updates the remaining weights of the network, and superior performance than Wanda which only takes into account the activation norm and weight magnitude
- Evaluated performance in different settings, including the N:M sparsity model supported by NVIDIA GPUs

## Weaknesses
- The impact of alpha in table 4 is rather weak. What's the rationale for choosing this alpha=0.5 when the gains are really small? Are there any other results that led to this decision?
- The details in table 6 are completely missing. Is this with any particular N:M sparsity pattern since there is a speedup in terms of runtime? The text as well as the table are very difficult to understand in the current form.
- If table 6 combines Wanda for attention layers, and DaSS for MLP layers, why aren't there any benchmark results reported in this setting? This seems like an unfair comparison.

---

> ### Author Response · Authors · 2024-11-18
> **Author Response to Reviewer  5FbZ**
>
> We first like to thank you for your time to review our work.
> Next, we would like to address all the weaknesses pointed by you point-by-point below:
>
> **W1:** The impact of alpha in table 4 is rather weak. What's the rationale for choosing this alpha=0.5 when the gains are really small? Are there any other results that led to this decision?
>
> In the ablation study, we show the results using Llama2-7B at 50% unstructured sparsity. Llama2-7B uses multi-head attention. In the Mistral-7B with grouped-query attention, the MLP module is larger, and the influence of α is more obvious.
>
> *Table: WikiText perplexity of 50% unstructured and 2:4 sparsity for Mistral-7B using different α*
> | α   | 50% sparsity| 2:4 sparsity |
> |-----------|---------| ---------|
> | 1.0     |  6.27  | 8.84 |
> | 0.5 | **6.15** | **8.39** |
>
> **W2:** The details in table 6 are completely missing.Is this with any particular N:M sparsity pattern since there is a speedup in terms of runtime? The text as well as the table are very difficult to understand in the current form.
>
> The sparsity pattern at table 6 is 50% and 70% unstructured sparsity pattern. The token generation speedup is tested using DeepSparse inference engine for CPU.
> LLMs utilize an auto-regressive framework in which tokens are produced sequentially. The token generation process is memory-bound. ​​Previous works, such as Wanda, focus on evaluating matrix multiplication speedups, but such metrics do not fully capture the actual inference speed improvements during token generation. To provide a more realistic benchmark, we utilized the DeepSparse inference engine, which is designed for CPU-based inference and is capable of leveraging unstructured sparsity effectively.
>
> As 2:4 sparsity for GPU, the most widely used LLM inference engine vLLM (https://docs.vllm.ai/en/latest/) is working for the integration of 2:4 sparsity. However, at this moment, its documentation has not yet provided guidelines on how to utilize 2:4 sparsity. We anticipate future updates will address this and enable a more thorough evaluation of GPUs.
>
> **W3:** If table 6 combines Wanda for attention layers, and DaSS for MLP layers, why aren't there any benchmark results reported in this setting? This seems like an unfair comparison.
>
> In Table 2 downstream tasks performance evaluation, we prune both the attention and MLP layers to better understand the performance gap between the dense model and the sparse model. Here, we prune the attention layers using Wanda, and prune MLP layers using DaSS.
>
> We sincerely hope our responses answer your questions, and we again take this opportunity to thank you for reviewing our work. We would be very happy to respond to any further questions you might have.

---

> > ### Comment · Reviewer_5FbZ · 2024-12-19
> > **Thanks for your response**
> >
> > I would like to thank the authors for all the provided clarifications. This addressed my concerns. While there are always areas for improvement, I believe this is a useful contribution.

---

### Comment · Action_Editor_dzm8 · 2024-12-22
**Request for inference speedup comparison**

Given that inference speedup is the main metric of interest in practice for compression methods, not showing any inference speedup gains severely limits the relevance of the contributions of this paper.
Can the authors compare the inference speedup achieved with DaSS to SparseGPT and Wanda, on both CPU and GPU, on the DeepSparse inference engine (as done in Table 6), for different levels of unstructured sparsity?

---

> ### Author Response · Authors · 2024-12-22
> **Response  to Action Editor dzm8**
>
> Dear Action Editor dzm8,
>
> Thank you for your question. We sincerely appreciate your dedication to reviewing our work, especially during this busy time of the year. We would like to address your concern as follows:
>
> In the field of unstructured pruning for LLMs, different methods are typically compared based on two main criteria: (1) task performance at the same sparsity level, and (2) pruning speed, as shown in Table 5. Regarding inference speedup or memory reduction, when evaluated under the same hardware and software environment, models produced by different pruning methods generally exhibit no statistical difference at the same sparsity level. As such, comparing different pruning methods in terms of inference speedup or memory reduction cannot yield meaningful results.
>
> For this reason, earlier works on LLM unstructured pruning, such as Table 5 in Wanda [1] and Table 7 in OWL [2], reported inference speedup without comparing different pruning methods. Moreover, since inference speedup has already been reported in earlier unstructured pruning studies and newer methods show no notable differences in this aspect, the inference speedup section has been widely omitted in more recent unstructured pruning research, such as [3]-[5]. Notably, [5] was recently published in TMLR.
>
> DeepSparse is an inference engine for CPUs that supports unstructured sparsity. As for inference speedup on GPUs, unstructured sparsity usually needs to be converted to an N:M sparsity pattern. Although [6] demonstrates speedup using several advanced N:M sparsities like 2:4, 32:64, and 16:64, its implementation has not yet been open-sourced, and currently, PyTorch only supports the 2:4 sparsity pattern. For Gate-Proj and Up-Proj in DaSS pruning, the 2:4 directions differ from those used in Wanda and SparseGPT. In the CUTLASS library, the pruning direction for 2:4 sparsity is fixed and cannot be modified. Conversely, the cuSPARSELt library supports transposition fusion, offering more flexibility. However, 2:4 sparsity is still a prototype feature in PyTorch, which leads to several limitations, including incomplete functionality and unresolved bugs. Furthermore, the usage of transposition fusion in cuSPARSELt within PyTorch has not been thoroughly documented. We anticipate that PyTorch will address these issues in future updates, improving the usability and robustness of the 2:4 sparsity feature. We have discussed the current software support for N:M sparsity in the limitations section.
>
> We sincerely appreciate the time and effort you've taken to review our paper. If you have further questions, we are more than happy to discuss them with you.
>
> Best regards,
> Authors
>
>
> **Reference:**
>
> [1]A Simple and Effective Pruning Approach for Large Language Models. ICLR 2024
>
> [2]Outlier Weighed Layerwise Sparsity (OWL): A Missing Secret Sauce for Pruning LLMs to High Sparsity. ICML 2024
>
> [3]Pruner-Zero: Evolving Symbolic Pruning Metric from Scratch for Large Language Models. ICML 2024
>
> [4]Discovering Sparsity Allocation for Layer-wise Pruning of Large Language Models. NeurIPS 2024
>
> [5]Fast and Effective Weight Update for Pruned Large Language Models. TMLR 2024
>
> [6]Sparse finetuning for inference acceleration of large language models. arxiv 2023

---

> > ### Comment · Action_Editor_dzm8 · 2024-12-22
> >
> > Thank you for your response.
> >
> > I am aware of the limitations you discussed of the software support for your method in N:M sparsity, that's why I asked for speedup results with unstructured sparsity.
> >
> > I still would like to see a comparison with the two baselines in terms of inference speedup on CPU if possible. This would be interesting to include, even if the results will be comparable. The fact that existing papers don't include such a comparison is  not necessarily a good example to follow.

---

> > > ### Author Response · Authors · 2024-12-24
> > > **Response to Action Editor dzm8 (2)**
> > >
> > > Thank you for your timely response!
> > >
> > > In response to your interest, we conducted an evaluation of the inference speed for different pruning methods under 50% and 70% sparsity. The results are shown in the table below:
> > >
> > > *Table: inference speed (tokens/second) for different sparse models as at 50% and 70% sparsity*
> > > | Model   | 50%| 70% |
> > > |-----------|---------| ---------|
> > > | SparseGPT | 5.64 | 8.23 |
> > > | Wanda |5.63 | 8.28  |
> > > | DaSS | 5.64 |  8.27 |
> > > | | |  |
> > > | Dense | 3.05 |
> > >
> > > As we can see, the inference speed differences between sparse models at the same sparsity level are negligible.
> > >
> > > We would like to emphasize that, similar to other unstructured pruning works, the primary contribution of our method lies in improving task performance at the same sparsity level and maintaining pruning speed comparable to Wanda for LLMs. The focus of our work is not on enhancing inference speed at the same sparsity level.
> > >
> > > If you have further questions, please do not hesitate to reach out.
> > >
> > > Best regards,
> > > Authors

---

### Decision · Action_Editor_dzm8 · 2025-01-01

**Recommendation:** Accept with minor revision

**Comment:**

The paper proposes a method, called DaSS, for pruning MLP layers in GLU-based LLMs. The proposed method assigns to each weight an importance score based on the product of its magnitude with the norm of the corresponding MLP intermediate activations, then prunes the same number of weights in each row or column of weights connecting to the same intermediate neuron. DaSS is a modification of the existing pruning method Wanda, where instead of using the input activation norm in the importance scores, it uses the intermediate MLP activation norm. This change is motivated by the dependency between weights connected to the same intermediate neuron. DaSS can also be adapted for the N:M sparsity pattern. The provided experiments show that DaSS outperforms SparseGPT & Wanda with N:M sparsity, in both perplexity and down-stream tasks accuracy, and in unstructured sparsity, in perplexity only on models with larger MLP layers, and in down-stream tasks accuracy only at high sparsity.

Strengths:
- The paper is well written and easy to follow
- The proposed method significantly outperforms Wanda in N:M sparsity in perplexity, and performs slightly better or comparably in other settings.
- The proposed method has the same low computational cost as Wanda.

Weaknesses:
- The proposed method is applying N:M sparsity column-wise for some of the weight matrices. However, current libraries supporting N:M sparsity either do not support column-wise layout or still have unresolved bugs with this feature. So the proposed method cannot be implemented for N:M sparsity with existing libraries. This limitation also prevents the authors from presenting inference speedup results on GPUs, which is the main metric of interest in practice.
- The proposed method only applies to GLU-based MLP layers.
- The proposed method only shows significant improvements over Wanda in the N:M sparsity setting, which is not supported with current libraries.

The official recommendations from the reviewers were mixed, with two reviewers recommending/leaning to accept and one reviewer leaning to reject. The authors have addressed in their responses some of the reviewers' concerns. The key concern raised by Reviewer GruC, and which I share, is the fact that the proposed method is not applicable in the N:M setting using current libraries, and this is the setting where the method shows significant improvements.
Nevertheless, this issue would likely be resolved in the near feature; the cuSPARSELt library will likely address the existing bugs in their support for column-wise N:M sparsity. I am thus recommending to accept the paper with minor revision.


Requested revisions:
- Make it clear earlier in the paper (in the introduction or even abstract) that the only change between the proposed method and Wanda is essentially the use of the intermediate MLP activation norm instead of the input activation norm in the importance scores.
- Highlight also in the introduction the limitation that existing libraries do not currently support the N:M sparsity variant of the proposed method.
- Adjust claims (again mostly in the abstract and introduction) that the proposed method outperforms baselines, clarifying that this is in terms of perplexity/accuracy, and not in terms of inference speed.
- Discuss the limitation that pruning model weights is not helpful in long-context inference, as in this setting the KV cache becomes the main bottleneck (from the response to Reviewer gEbX).
- Include the explanation of the difference in the PPL values you get for Wanda from the original paper (from the response to Reviewer GruC).
- Discuss the related work "Fast and Effective Weight Update for Pruned Large Language Models"
- Explain the intuition/motivation behind using $\alpha < 1$ in Gate-Proj and Up-Proj, and $\alpha = 1$ in Down-Proj. Include ablation experiments for having a hyper-parameter $\alpha$ in Down-Proj too.
- Include results for $\alpha = 0$ in Table 4 and $\alpha \in  \\{0, 0.25, 0.75\\}$ in Table 11.
- Include the CPU inference speedup comparison results provided in your response to my comment.

**Audience:**

yes but limited. The proposed method only shows significant improvements over the existing pruning method Wanda in the N:M sparsity setting, where the method is not supported with current libraries.

**Claims And Evidence:**

Some claims are not well supported (see comment for more details):
- the claim that the proposed method is novel is a bit misleading, since the method is a simple modification of the existing pruning method Wanda (Sun et al., 2024). This should be clarified.
- the claim that the proposed method can be easily extended to hardware-friendly N:M sparsity patterns is not fully accurate, since it cannot actually be implemented with existing libraries supporting N:M sparsity.
- the claim that the proposed method outperforms baselines should specify that this is only in terms of perplexity/accuracy and not in terms of inference speedup.

---

> ### Author Response · Authors · 2025-01-20
> **Camera-ready Version Submitted**
>
> Dear Action Editor dzm8,
>
> First and foremost, thank you very much for your decision to accept our work with minor revision. We sincerely appreciate the detailed and thoughtful feedback you have provided, which has been invaluable in improving our work. We are also deeply grateful for your effort in reaching a decision during such a busy period, and we truly value your timely and careful consideration.
>
> Since we are unable to use color annotations in the camera-ready version, we have outlined below the specific locations corresponding to each requested revision:
>
> - (1) We added this in the Page.3 Introduction Part: DaSS refines Wanda’s pruning criterion by replacing input activation norms with those of the intermediate activations generated within the MLP. Specifically, while Wanda’s importance score …
> - (2) We added this to the Page.3 introduction part: However, realizing the full performance benefits of this aspect is currently constrained by …
> - (3) We adjusted this in the abstract and Page.3 introduction, and added footnote 2 for further explanation.
> - (4) We added this on Page.11 the last paragraph of Limitation part.
> - (5) We explained this on Page.6 footnote 3 and the related part of footnote 3.
> - (6) We discussed this on Page.10 Pruning LLM. part
> - (7) We updated the text on Page.5 between Eq. (4) and Pruning Granularity part. We added experiments on Page.17 Table 12.
> - (8) We updated Table 4 and Table 11.
> - (9) We added it on Page.17 A.6 Inference Speed Comparison, and added such explanation on Page.9 Inference efficiency part.
>
> We hope these revisions address the concerns and suggestions raised in the review process. Please feel free to let us know if there are any additional changes or clarifications required. Once again, we sincerely thank you for your constructive feedback and support throughout the review process.
>
> Best regards,
>   Authors